# Systematic review of educational interventions to improve the menstrual health of young adolescent girls

Rebecca Lane Evans ,[1] Bronwyn Harris,[1,2] Chinwe Onuegbu ,[1] Frances Griffiths [1,2]

[1]Division of Health Sciences, Warwick Medical School, University of Warwick, Coventry, UK
[2]Centre for Health Policy, School of Public Health, University of the Witwatersrand, Johannesburg, South Africa

**Correspondence to**
Ms Rebecca Lane Evans;
rebecca.r.evans@warwick.ac.uk

## ABSTRACT

**Objectives** To systematically review interventions that include an element of menstrual education delivered to young adolescent girls.

**Design** This was a systematic review and meta-analysis. Selected articles were quality assessed using the Mixed Methods Appraisal Tool quality appraisal checklist. A meta-analysis was conducted on a subset of articles, and the effect size of the intervention was calculated using Cohen's d. A logic model was constructed to frame the effect of menstrual education interventions on menstrual health.

**Setting** Papers reporting on interventions in high-income and low-income and middle-income countries were sought.

**Information sources** Seven electronic databases were searched for English-language entries that were published between January 2014 and May 2020.

**Participants** The interventions were aimed at younger adolescent girls aged 10–14 years old.

**Interventions** The interventions were designed to improve the menstrual health of the recipients, by addressing one or more elements of menstrual knowledge, attitude or practices (KAP).

**Eligibility criteria** Interventions that had not been evaluated were excluded.

**Primary and secondary outcomes** The most common type of output was a difference in knowledge or skill score ascertained from a pre and post test. Some studies measured additional outcomes, such as attitude or confidence.

**Results** Twenty-four eligible studies were identified. The number of participants varied from 1 to 2564. All studies reported improvements in menstrual KAP. The meta-analysis indicates that larger effect sizes were attained by those that encouraged discussion than those that distributed pamphlets.

**Conclusions** Education interventions are effective in increasing the menstrual knowledge of young adolescent girls and skills training improves competency to manage menstruation more hygienically and comfortably. Interactive interventions are more motivating than didactic or written. Sharing concerns gives girls confidence and helps them to gain agency on the path to menstrual health.

**Trial registration number** For this review, a protocol was not prepared or registered.

### STRENGTHS AND LIMITATIONS OF THIS STUDY

⇒ This review is of the international literature including high-income and low-income and middle-income countries.
⇒ It uses mixed methods approaches and synthesises both quantitative and qualitative studies.
⇒ A logic model has been constructed to frame the effect of menstrual education interventions on menstrual health.
⇒ There may be publication bias in the reporting of positive outcomes only.
⇒ A full meta-analysis was not possible due to heterogeneity in methods.

## INTRODUCTION

Globally, young adolescent girls are ill-prepared for menarche and menstruation.[1–4] In many cultures, menstruation is a taboo subject[5] and many girls are ignorant of it until they start bleeding.[6] Negative experiences of menarche and early menstruation can cause poor menstrual health.[2 7]

'Menstrual health' is an emerging area of health research. It is a broad term that encompasses the hygienic management of menstruation and the psychological components of well-being such as confidence, dignity and self-esteem.[8 9] It is an expansion of the concept of menstrual hygiene management (MHM), mostly used in low-income and middle-income country (LMIC) contexts to describe the challenges of hygienically managing menstruation with a lack of resources, especially pads, water and soap.[10–12] The use of dirty rags to absorb menstrual blood has been proposed as a cause of reproductive tract infections (RTIs) and cervical cancer,[13–15] as infectious agents may be introduced into the reproductive tract from such materials. Even clean materials that are not changed regularly may smell[3] and some materials can cause chafing and irritation. Without sufficient water to wash the blood from their genitals or their hands,[16] girls can become

uncomfortable and anxious.[17] Several studies in LMIC have shown that menstruation is associated with a reduction in participation in activities and an increase in school absenteeism.[14 18 19] In high-income countries (HICs), 'period poverty' has only recently been recognised as an issue for certain groups, such as homeless women,[20] but a whitepaper by PHS Group, UK (a hygiene services provider in the UK) suggests period poverty is more wide-reaching, particularly among school girls, and is a factor contributing to menstrual anxiety and school absenteeism.[21] Significantly less research has been conducted in HICs around menstruation and participation, although it is widely recognised that girls dropout of sports activities around the age of puberty.[22 23]

Discussing menstruation is almost universally a taboo, making it difficult for girls to learn about it and know what is normal or when to seek help. A study by Gultie *et al*[24] into menstrual knowledge of adolescents in Ethiopia found that 33% of participants never talked about it with anyone.[24] In Jordan, a phenomenological analysis of experiences of menarche reported that girls believed talking about menstruation was 'rude'.[25] A study into puberty communication in the Czech Republic and China found that both men and women were complicit in perpetuating menstrual stigma, with mothers instructing their daughters to keep their menstrual status 'secret'.[26] In India, Rani *et al* found that although 61.3% of adolescent girls suffered from debilitating dysmenorrhea, they felt they were expected to 'tolerate' it as a natural process and only 1.6% had ever consulted a physician.[13]

Menstrual knowledge usually comes from mothers[7] but a number of studies have shown that the knowledge of mothers themselves may be incomplete[27 28] and they may actually perpetuate cultural myths and misinformation.[29 30] In HICs, such as Australia and the UK, menstruation may be taught at school as part of sex education, if it is mentioned at all, but many girls do not get the practical information that they need.[31 32] In both HIC and LMIC, non-government agencies have stepped in to try to plug the gap by providing menstrual education interventions.

In this review, our objective was to describe and evaluate the impact of menstrual education interventions intended to equip young adolescent girls with the knowledge and skills to promote menstrual health.

Two previous reviews of papers published prior to January 2015 focused on the more narrow 'MHM' in LMIC.[14 33] The term menstrual health is now preferred to hygiene, partly to avoid the suggestion that menstruation *per se* is unclean, but mostly to emphasise the holistic nature of the menstrual experience. This literature review only includes publications since 2014 in order to capture that change.

## METHODS

This is a systematic review of the published literature of interventions that include an element of menstrual education delivered to young adolescent girls. We report according to the Preferred Reporting Items for Systematic Reviews and Meta-Analyses guidelines.

### Patient and public involvement

No patient involvement.

### Publication date and language

This review draws on papers published from January 2014 until May 2020 to bring the field up to date. Only reviews published in the English language have been included.

### Participants

For this review, we were interested in interventions targeted at young adolescent girls aged 10–14 years old.

### Inclusion and exclusion criteria

Studies of interventions that had been evaluated were sought.

All interventions that had a component of menstrual information transfer were included if the intervention sought to:
► Increase knowledge of menstruation to reduce anxiety and shame, and normalise the experience.
► Increase skills and competencies to manage menstruation comfortably and hygienically.
► Increase awareness of strategies for self-care of menstrual symptoms.

Interventions that were straightforward 'menstrual education' in which lessons about puberty, anatomy and hygiene were delivered by teachers or other educators, to both boys and girls, were eligible. Interventions focused on skills training, such as correct menstrual cup insertions, and delivered by nurses or other key workers were also eligible. Programmes that facilitated peer or self-guided learning through the provision of resources or spaces (both physical and remote) for learning to occur were also eligible for inclusion.

Studies were excluded if the improvements were in hardware such as toilets or pads without any accompanying education or training, or if they provided hygiene education without reference to menstruation specifically. Studies were excluded if they were about abnormal menstruation, menstrual problems as a comorbidity, or if the research was investigating endocrinology or non-human models. Studies that described existing knowledge, attitude or practices (KAP) without any intervention were also excluded.

Studies that included adolescents up to the age of 19 were not excluded if the aim were to instruct menstruators with limited experience of menstruation. Some studies included older girls because they were intellectually disabled and were part of the intervention based on intellectual age rather than chronological age. Some studies included older girls because they were members of classes assigned by grade rather than age. Studies about interventions aimed at adult women were excluded.

Study design and quality were not part of the criteria to capture as broad an interpretation of menstrual education as possible.

**Table 1** Search terms used

| Search terms used |
| --- |
| Search term 1<br>'adolescen* OR girl* OR teenage* OR youth* OR young OR pre-adolescen* OR school-girl OR Out-of-School-Youth OR OOSY OR female OR woman<br>AND Search term 2<br>Menstrua* OR menarche* OR mense* OR catamenia OR menarche* OR menstrual health OR menstrual hygiene OR menstrual management OR sanitation OR menstrual etiquette<br>AND Search term 3<br>know* OR understand*OR manage* OR learn* OR apprehen* OR comprehensi* OR educat* OR aware* OR familiar* OR proficien*<br>AND Search term 4<br>AND arrangement* OR evaluat* OR initiative* OR intervention*OR model* OR package* OR pilot* OR program* OR project* OR provision* OR regime* OR scheme* OR strateg* OR trial* OR approach* OR polic* |

| Database returns | | |
| --- | --- | --- |
| Database | Search terms and filters | Returns |
| ASSIA (proquest) | 1 (ab), 2 (ti), 3 (ab), 4 (ab)<br>01/01/14 to 01/09/19 | 15 |
| CINAHL | 1 (ab), 2 (ti), 3 (ab), 4 (ab)<br>01/01/14 to 01/09/19 | 126 |
| EMBASE<br>Including MEDLINE | 1 (ab), 2 (ti), 3 (ab), 4 (ab)<br>2014–2019<br>English, Female | 732 |
| Sociological abstracts (proquest) | 1 (ab), 2 (ti), 3 (ab), 4 (ab)<br>01/01/14 to 01/09/19 | 14 |
| Web of Science | TS=(1), (3), (4) AND Ti=(2)<br>Last 5 years | 323 |
| IBBS (proquest) | 1 (ab), 2 (ti), 3 (ab), 4 (ab)<br>01/01/14 to 01/09/19 | 26 |
| TRoPHI (Eppi centre, google) | No wild cards<br>Title<br>Menstruation and Knowledge<br>2014–2019 | 4 |
| TOTAL SAVED to EXCEL | | 1240 |

## Protocol for identification of academic literature

### Screening

As the field of menstrual health education is highly interdisciplinary, we searched key medical and social science databases: ASSIA Applied Social Science Index and Abstracts; CINAHL Cumulative Index to Nursing and Allied Health Literature; EMBASE Excerpta Medica database; MEDLINE Medical Literature Analysis and Retrieval System Online; Sociological Abstracts; Web of Science; IBBS International Bibliography of the Social Sciences; TRoPHI Trials Register of Promoting Health Interventions.

The search parameters combined the target population, menstruation, education and programme (table 1).

Two reviewers (RLE and CO) screened abstracts and full texts of all citations obtained for eligibility independently. Data extraction of eligible material and the quality assessment was conducted by RLE using a data extraction framework agreed on by FG, BH and RLE. The quality assessment tool was agreed on by RLE, FG and BH.

RLE used the Mixed Methods Appraisal Tool (MMAT) to assess the quality and this was verified by FG and BH.

### Outputs and data extraction

The outputs of menstrual education are changes to menstrual KAP. In quantitative studies, preintervention and postintervention KAP scores were collected, as well as data to support the validity of the scores, such as sample frame and number of participants. Other descriptive measures used in qualitative studies and contextual factors that might have a bearing on the outputs were also recorded where given.

### Quality assessment

We used the MMAT for quality assessment because it is suitable for a variety of study designs. The MMAT components focus on the clarity of the research questions and the appropriateness of the data collection methods. Our intention was to consider the MMAT assessment in the interpretation of study findings.

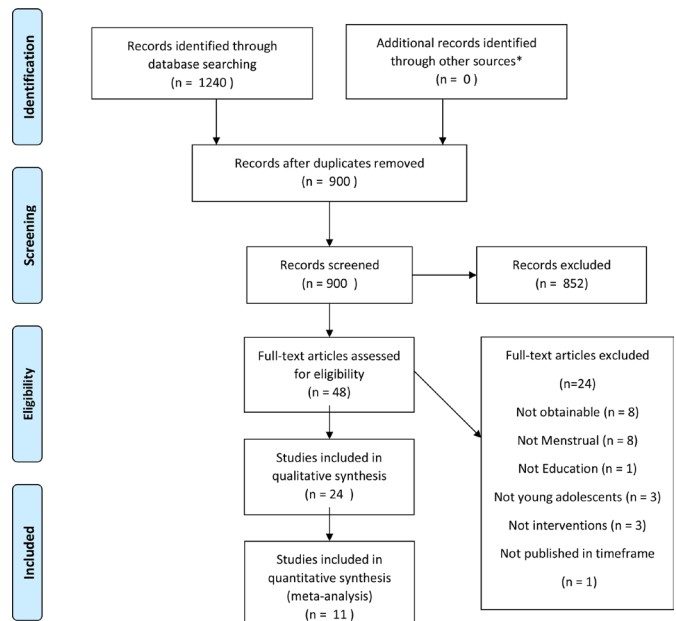

**Figure 1** Preferred Reporting Items for Systematic Reviews and Meta-Analyses flow-chart. *Grey literature.

## Analysis and synthesis
### Quantitative analysis
The results of quantitative studies were tabulated for comparison. Not all data could be converted but for studies that reported quantitative results with a pre-test and post-test score of menstrual knowledge, we found the effect size by calculating the standard mean difference using Cohen's d.Effect sizes can lie between 0 and infinity. Cohen suggested that a small effect size is a value of 0.2, a medium one is 0.5 and a large one is 0.8. Some social science disciplines report much larger sizes and the statistical guidance was revised to suggest that a medium effect is the average of those in the relevant literature.[34 35] However, Cohen's d has not previously been calculated for this discipline. We found the average of those in this review and ranked the results accordingly.[36]

### Qualitative analysis
A qualitative comparative analysis[37] was made of all studies by scrutinising the data extracted for common themes and reporting them in a narrative summary.

### Synthesis: building a logic model
Logic models enable the key findings of the analysis to be synthesised into a theory of change. We constructed a logic model to frame the effect of menstrual education interventions on menstrual health.

## RESULTS
A total of 1240 papers were recovered using the search terms. 900 remained after the removal of duplicates and 48 were saved to Excel for full-text screening. Overall,

852 did not meet the inclusion criteria. This was largely because they were not interventions but studies of menstruation (figure 1).

## Study characteristics
Twenty-four papers met the inclusion criteria: they were grouped into the following categories of study design according to the MMAT Quality Appraisal tool: 12 Randomised Controlled Trials (RCTs); five quantitative descriptive studies; one quantitative non-random study; five mixed methods studies and one qualitative study. See online supplemental file 1 for further detail on study design.

The study dates ranged from 2012 to 2017, with dates of publication ranging from 2014 to 2019. Studies were undertaken in Iran (six), Turkey, Indonesia (two), Ethiopia, India (four), Bangladesh, Uganda (four), the USA, Nepal, Kenya (two) and China.

Of the studies, two included boys,[9 19 38] one included mothers[28 30] and two focused on intellectually disabled adolescent girls.[39 40] The number of study participants varied from 1 to 2564. The total number of participants in the 20 different intervention studies was 10 362.

The amount of time after the completion of the intervention to the assessment varied from the same day to up to 5 years, with 13 of the studies in the range of 1–9 months and the mode being 1 month. A common theme was allowing one menstrual cycle to pass before retesting.

## Delivery
Eight of the interventions used health professionals as instructors.[6 8 9 30 39 41–44] Five were researcher led,[30 40 45–47] five were led by teachers,[19 48–51] two used peer educators[8 42] and the others did not make it clear. A wide range of intensity and duration models were used, ranging from three 5 min videos in the space of one afternoon[38] to 1 hour per day for 4 months.[41] Only two embedded menstrual education into the school curriculum.[50 51]

## Quality assessment
The methodological quality of study designs was mixed: 11 were rated as high quality and 13 as moderate to low. Those considered to be of the highest quality were RCTs, which included comparison groups. Some of the studies (nine) did this at the whole-school level, which is recommended in educational interventions to prevent contamination of the intervention group with the control.[52] The research questions were clear and the data collection methods appropriate. Of the other studies, several methodological limitations were noted; commonly, neither the delivery team nor the participants were blinded (9); adequate randomisation of the participants was lacking (10) and/or relevant confounds were not identified or controlled (4). The quality of data analysis also varied considerably, with the weakest having small sample sizes and no measure of statistical significance (two).

## Interventions

There was a range of intervention types. Eight of the interventions employed traditional education in the form of didactic teaching, sometimes supplemented with posters, flip-charts and question-and-answer sessions.[18 41 43 46 47 50 51 53] Two interventions employed a more formal lecture presentation and gave out some supporting literature.[30 45] Two interventions used stories and video presentations[38 40] and two interventions distributed puberty books[4 48] without further teacher input. Three interventions facilitated learning through peer-education[8 42 54] and seven different interventions focused on MHM training: some demonstrated pad usage with a menstrual kit[9 19 40 49] or using a doll[39] and two instructed participants on the use of menstrual cups.[44 55]

## Aims of the studies

The common aim of the studies was to evaluate the impact of a menstrual education intervention on menstrual health for adolescent girls. A number of studies were more broadly about puberty education.[30 38 45 46 48 50] Several studies measured menstrual KAP.[4 6 30 38 41 42 47–49 53 54] Nearly all of the studies used a pretest, post-test model but one study used a post-test only model.[38] A small number specifically focused on MHM and evaluated training on pad replacement or cup insertion.[9 19 33 39 40 42 43 55 56]

## Analysis

The quantitative and qualitative results are reported under four main themes: menstrual knowledge, menstrual attitudes, menstrual practices and multicomponent interventions.

### Quantitative results
#### Menstrual knowledge

A meta-analysis was conducted on 11 studies, which measured a change in menstrual knowledge following an intervention. A visual inspection of forest plots showed that all studies found a significant improvement in menstrual knowledge. Where studies reported the mean and SD of a menstrual knowledge questionaire, we calculated the effect size using Cohen's d (table 2).

The average effect size of studies in this review was 3.44. Taking this as a medium effect size, we ranked them lowest—highest and suggest that <2 is low and >5 is high. Where we could not calculate an effect size, we have calculated % change in score. Due to the limited and heterogeneous nature of the data, we interpret the results only relative to the other studies in this review.

The effect size of those that distributed pamphlets and books was lowest at 0.33,[30] followed by those that showed videos 1.40[40] and then lectures with question and answer sessions 2.13[47] and 4.81.[53] Small group or peer teaching was high at 5.337 and 10.044, respectively. Large effect sizes may occur due to small sample sizes.

#### Menstrual attitudes

Five studies measured menstrual attitudes.[4 38 45 47 49]

Four interventions reported significantly different (p<0.05) attitude scores, and three of those provided pamphlets that addressed cultural restrictions.[4 45 49] The other was an intervention on dysmenorrhea and self-care and included pamphlets with video and peer sharing. Girls who had taken part had a significant increase in confidence and decrease (p<0.001) in 'bothersome' menstrual attitude.[47] The only intervention that did not find a significant difference in attitude pretest and post-test involved puberty education videos shown to early adolescent boys and girls.[38]

#### Menstrual practice

An intervention that trained intellectually disabled adolescents in an 18-item pad replacement skill set found that pretraining and post-training differences were statistically significant (p<0.001).[39] A feasibility trial into the use of the menstrual cup by school girls in Kenya[44] found that usage increased as time went on and culminated in 96% usage after 9 months. There was also an increase in hygiene, with the menstrual cup reported as reducing the prevalence of Sexually Transmitted Infections (STIs) from 19.2% to 12.9% (p=0.018).[55]

#### Multicomponent interventions

An education intervention in India was part of a bigger project that involved pad provision and improved sanitation in schools. After 4 years, compared with unimproved schools, school attrition had fallen from 11% to 6% (p<0.003).[51] The effect of menstrual education alone cannot be separated out.

### Qualitative results
#### Menstrual knowledge

All studies reported an increase in menstrual knowledge. Interventions that used peer education and group counselling[8 42] were as effective as those delivered by medical professionals.[49]

Those interventions that had a degree of interactivity were more effective than those that only gave out information. Those that encouraged discussions found that they led to an increased willingness to talk about menstruation and a greater awareness of what is normal.[8 46]

Only two interventions embedded menstrual education into the school curriculum.[50 51] In Nepal, some schools had received the Water, Sanitation and Yygiene (WASH) in Schools programme.[57] However, the girls were highly critical of their teachers, especially male teachers. They complained that 'teachers often got embarrassed, referred students to their textbook, and did not answer questions'.[50]

#### Menstrual attitudes

Most studies commented on an improved menstrual attitude and one noted a reduction in anxiety.[46] More than one study noted an improvement in confidence in performing menstrual healthcare behaviour, such as

**Table 2** Effect size of interventions designed to improve menstrual knowledge

| Study type and measure | First author and date of publication | Intervention | Number of questions/N | Reliability/ Cronbach's alpha (except where stated otherwise) | Sample size/n | Treatment | Mean/arbitrary units (except where stated otherwise) | SD/arbitrary units (except where stated otherwise) | Number of individuals scoring>75% (good) | % change in number of individuals scoring>75% | Statistical test/p value | Effect size/ Cohen's d/ rank and impact |
|---|---|---|---|---|---|---|---|---|---|---|---|---|
| Control and intervention pretest, post-test | Blake et al 2018 | Distribution of puberty education books in the local language | 9 | 0.77 | 318 | Control | (Mean difference=0.18) | (Pooled SD=1.4) | No data | | Wald χ², <0.001 | 0.79 Second low |
| | | | | | 318 | Intervention | (MD=1.06) | (Pooled SD=1.52) | | | | |
| | Jarrahi et al 2020 | Small group and peer teaching | 34 | 0.78 | 30 | Control pre | 45.1 | 8.4 | No data | | Kruskal-Wallis test,<0.001 | |
| | | | | | 30 | Control post | 52.2 | 1 | | | | |
| | | | | | 30 | Intervention 1 pre (small group) | 48.3 | 6.1 | | | | Small group 5.34 Eighth high |
| | | | | | 30 | Intervention 1 post | 84.5 | 8.5 | | | | |
| | | | | | 30 | Intervention 2 pre (peer) | 44.1 | 1.7 | | | | Peer group 10.04 Nineth high |
| | | | | | 30 | Intervention 2 post | 93.3 | 5.7 | | | | |
| | Setyowati et al 2019 | Booklet distribution | 14 | 0.886 | 87 | Control pre | No data | No data | No data | No data | χ² <0.001 | (Medium) |
| | | | | | 87 | Control post | | | 17.2 | | | |
| | | | | | 87 | Intervention pre | | | 58.6 | 54.95 | | |
| | | | | | 87 | Intervention post | | | 90.8 | | | |
| | Sharma et al 2015 | Interactive teaching programme led by school nurses | 15 | Test, retest r=0.93 | 25 | Control pre | 8.02 | No data | No data | | T-test<0.001 | 4.48 Sixth medium |
| | | | | | 25 | Control post | 8.06 | | | | | |
| | | | | | 25 | Intervention pre | 8.04 | | | | | |
| | | | | | 25 | Intervention post | 12.6 | | | | | |
| | Su et al 2016 | Lecture, question and answer session | 13 | KR20=0.64 | 56 | Control pre | 5.5 | 2.54 | No data | | T-test<0.001 | 2.13 Fifth medium |
| | | | | | 56 | Control post | 5.71 | 2.3 | | | | |
| | | | | | 60 | Intervention pre | 5.73 | 2.56 | | | | |
| | | | | | 60 | Intervention post | 10.22 | 1.92 | | | | |
| | Valizadeh et al 2017 | Lecture, booklet and pamphlets | 15 | 0.72 | 120 | Control pre | 8.5 | 2.5 | No data | | General linear model <0.002 | 0.33 First low |
| | | | | | 120 | Control post | 9.1 | 2.4 | | | | |
| | | | | | 124 | Intervention pre | 8.2 | 2 | | | | |
| | | | | | 124 | Intervention post | 9.8 | 1.8 | | | | |

Continued

**Table 2** Continued

| Study type and measure | First author and date of publication | Intervention | Number of questions/N | Reliability/Cronbach's alpha (except where stated otherwise) | Sample size/n | Treatment | Mean/arbitrary units (except where stated otherwise) | SD/arbitrary units (except where stated otherwise) | Number of individuals scoring>75% (good) | % change in number of individuals scoring>75% | Statistical test/p value | Effect size/Cohen's d/ rank and impact |
|---|---|---|---|---|---|---|---|---|---|---|---|---|
| | Kheirollahi et al 2017 | Lecture, question and answer session | 23 (100 point scale) | 0.8 | 76 | Control pre | 52.58 | 6.58 | No data | | T-test<0.001 | 4.81 Seventh medium |
| | | | | | 76 | Control post | 52.77 | 6.87 | | | | |
| | | | | | 76 | Intervention pre | 55.83 | 6.77 | | | | |
| | | | | | 76 | Intervention post | 86.36 | 7.11 | | | | |
| Control and intervention post-test only | Hurwitz et al 2017 | Health education videos, using animation | 27 | 0.72 | 40 | Control | 11.27 | 3.73 | No data | | T-test<0.001 | 1.40 Third low |
| | | | | | 40 | Intervention | 15.67 | 2.4 | | | | |
| One group pretest, post-test | Arasteh et al 2019 | Group counselling and pamphlets | 15 | 0.8 | 30 | Pre | 6.8 | 3.32 | No data | | $\chi^2$ <0.001 | 1.64 Fourth low |
| | | | | | 30 | Post | 11.3 | 2 | | | | |
| | Haque et al 2014 | Training with field manual by medical professionals | 10 | 0.73 | 416 | Pre | No data | No data | 51 | 61.57 | $\chi^2$ <0.05 | (High) |
| | | | | | 416 | Post | | | 82.4 | | | |
| | Chadawada et al 2017 | Didactic teaching with posters and videos | 4 | No data | 250 | Pre | | | 72.7 | 19.12 | $\chi^2$ <0.05 | (Low) |
| | | | | | 250 | Post | | | 86.6 | | | |

requesting pain relief for dysmenorrhea.[47] Some studies observed an increased confidence of girls to push back against cultural restrictions, or harmful practices.[8 18 46 49 58]

### Menstrual practices

Skills are required to use pads and cups so that they are positioned correctly, are comfortable and do not leak. The cup feasibility trial in Kenya found that on-going training and support may be required to master the technique over a period from 6 months to 1 year.[44 55] Education was also found to be an important component of skill acquisition in Uganda, where pad provision accompanied by education was shown to be more effective than pad provision alone.[18]

### Multicomponent interventions

The Menstrual Health Intervention and School Attendance in Uganda (MENISCUS) programme attributed its success to the synergy of five combined elements; teacher training on puberty education, a drama skit, pad provision, pain relief provision, and WASH facility improvements.[9]

## Synthesis: the logic model

Guided by logic models developed for school-based interventions,[59] we worked backwards from the higher order aim of good menstrual health to propose a chain of causal events.

The aim of good menstrual health is the distal outcome to the intervention. It is characterised by girls feeling empowered and having agency to make choices about their own bodies and lives. They can choose a suitable menstrual product to meet their individual needs. They track their menstrual cycle to be well-prepared so that they are not caught out at school and have to go home, and they engage as necessary with reproductive health services, without shame or stigma. Girls that have agency are able to control their menstruation and not the other way around. They can focus on their school work and reach their potential.

Preceding the distal outcome is the intermediate outcome; unrestricted mobility and participation. Girls should be able to carry out normal activities such as eating/drinking with the family, attending school and playing sport when they are menstruating. This requires confidence in their own ability to manage outside of the confines of the house, and determination to enter spaces from which they are traditionally excluded.

Below that is the proximal outcome; hygienic and comfortable menstruation management. Girls should be able to use suitable menstrual products. They should be able to use water and soap to clean away menstrual blood, and they should be able to practise self-care to relieve the symptoms of dysmenorrhea, such as yoga. If they need painkillers or a rest, they should be able to request them of parents and teachers without embarrassment.

Menstrual KAP underpins these outcomes. In a theory of change, girls require knowledge of the menstrual cycle

to prepare products. They may need skills to use the products correctly to avoid the risk of discomfort, leaks or of contracting RTIs. They may need confidence to ask for products and services. They may need awareness of self-care practices. They should know what is normal and when to seek help.

## Inputs and outputs to the logic model

Menstrual education and training are the inputs. The output is improved menstrual KAP. The outputs are linked to the outcomes.

The results of the review provide evidence that menstrual education improves the menstrual KAP of young adolescent girls. It is suggested that increasing the menstrual KAP of girls increases their confidence to seek further knowledge and skills in a positive feedback loop (see figure 2). Menstrual education is viewed as underpinning the logic model and is the first step to achieving menstrual health.

## DISCUSSION

All 24 included studies that evaluated some form of menstrual education intervention reported that there was a measurable improvement in the menstrual KAP of young adolescent girls. Menstrual experience and need may be the motivators, as there was little improvement in the knowledge or attitude of very early adolescent girls or boys.

Most of the interventions were delivered in the school setting where it is relatively easy to reach the target group, although only two embedded the education into the school curriculum. In theory, schools should have good coverage and objectivity for delivery of this sensitive information at an appropriate time. However, teachers themselves may be ill-equipped to teach about menstruation without proper training.[60]

A larger effect was gained with the more interactive interventions that included question and answer sessions. We suggest that this relates to the higher degree of participation, and concurs with current educational philosophy about the importance of active learning,[61] based on constructivist theory.[62 63] Gardner added to Dewey's early work on active learning when he described 'transformative' teaching. This involves using a range of methods that encourage the learner to find their own entry point and engage with the subject, often using space and creativity, and linking with their own experience.[64] Discussing menstruation gives the girls agency to determine what it is that they need to know for themselves.

Those interventions that demonstrated skills and allowed for physical touch were also very effective. Other hygiene interventions that have been evaluated have pointed to the positive impact of a physical interaction with the tools of behaviour change.[65]

A logic model has been constructed to frame the effect of menstrual education interventions on menstrual health. Menstrual education is seen as underpinning all

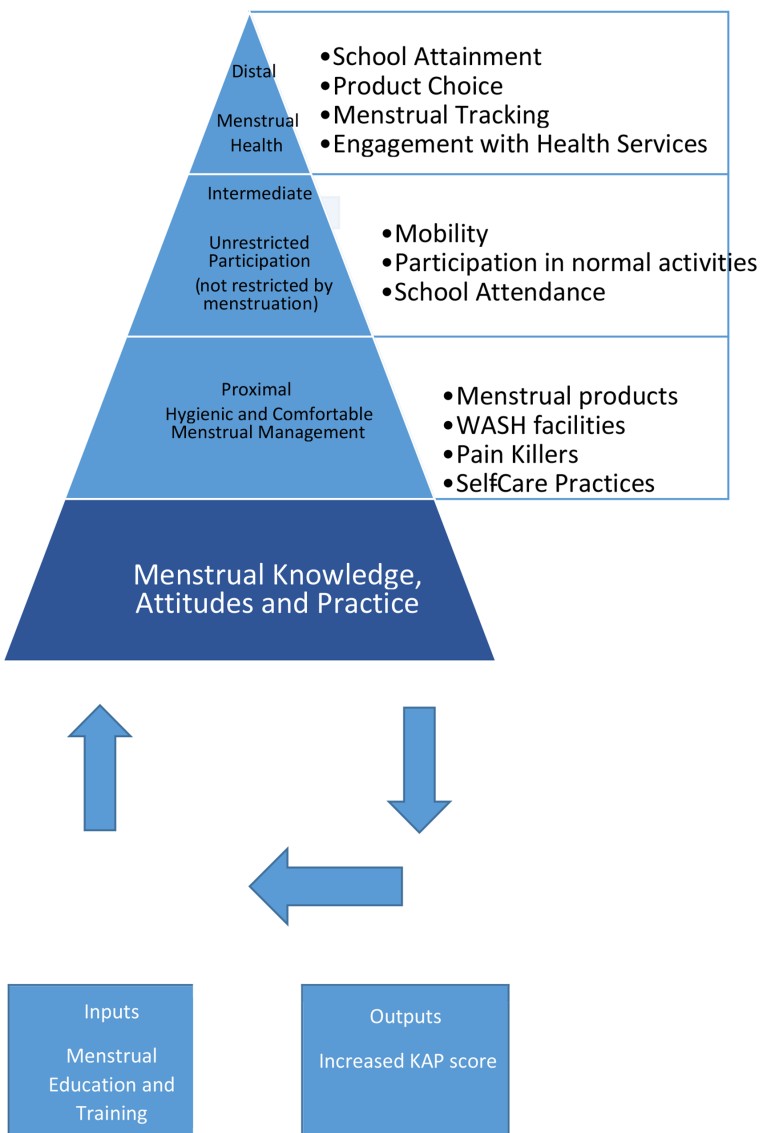

**Figure 2** The logic model. KAP, knowledge, attitude and practices; WASH, water, sanitation and hygiene.

desirable outcomes and programmes can be optimised by including an education aspect. There was evidence that interventions were successful in normalising menstruation and improving menstruation management, which is the proximal outcome of the logic model. There was more limited evidence for an improvement in school attendance and mobility or a refusal to accept menstrual restrictions. These are intermediate outcomes and it might be expected to take some time to move from the proximal to the intermediate outcomes in the theory of change.

Multicomponent interventions may be more successful than single components in achieving the distal outcome of menstrual health and well-being. Girls need an enabling environment as well as knowledge. From a constructivist perspective which places learning within a social context[63 66] interventions that seek to improve the menstrual literacy of the whole community and reduce menstrual stigma may be more effective in achieving the macrodistal outcomes of girl empowerment and gender equality.

## Implications for policy and practice

This review provides evidence that menstrual education has a positive effect on the menstrual KAP of adolescent girls and needs to be delivered by trained personnel who are confident to lead discussion. Especially but not exclusively in LMIC, where resources are limited, it would be prudent to ensure that menstrual education is embedded into the school curriculum and that teachers receive specialist training.

Progress towards menstrual health is limited without an enabling environment. In order to achieve the more distal outcomes of the logic model, programme and policy-makers need to address the menstrual literacy of the wider population. Multicomponent interventions that speak to different actors and include hardware and software provision alongside menstrual education may make menstrual health more attainable.

## COVID-19

This review was carried out on studies conducted before the global pandemic began in March 2020. The subsequent lockdown has had a profound effect on education, and many programmes have had to go on-line. We would encourage menstrual educators to be mindful of the benefits of interaction and make use of on-line teaching platforms that facilitate discussion in break-out rooms.

## Limitations of this review

The review was carried out in the English language, which may have missed some publications. Because menstrual health is an emerging topic with evolving terminology, search terms may not have adequately captured all currently used descriptors.

As a mixed methods review, there are a number of systemic limitations derived from comparing heterogeneous data sets. The studies did not measure the same outputs and the methodological quality of the studies was mixed. Two were aimed at the intellectually disabled. It is possible that the level of menstrual knowledge was so low at baseline that any educational intervention would result in an improvement.

Although all interventions reported positive outcomes, this may be due to publication bias, where only significant results are shared. Cohen's d has not previously been calculated for this discipline and therefore the magnitude of the effect size can only be considered relative to others in this review.

The number of studies was small, and only one study was from an HIC, so it is difficult to say how applicable the conclusions are to an HIC. More research needs to be done in this area, particularly as period poverty has been increasingly reported since the start of the pandemic in HIC.

**Contributors** RLE, FG and BH designed the review. RLE and CO reviewed titles, abstracts and full texts for eligibility. Disagreement was resolved by discussion and where necessary FG and BH offered their view. RLE, FG and BH agreed on a data extraction framework, which was then carried out by RLE. The quality assessment tool was agreed upon by RLE, FG and BH. RLE used the Mixed Methods Appraisal Tool to assess the quality and this was verified by FG and BH. RLE prepared the manuscript and it was reviewed and edited by FG and BH.

**Funding** This work was supported by the National Institute for Health Research Unit on Improving Health in Slums. Funder reference: 16/136/87.

**Competing interests** None declared.

**Patient and public involvement** Patients and/or the public were not involved in the design, or conduct, or reporting, or dissemination plans of this research.

**Patient consent for publication** Not applicable.

**Ethics approval** Not applicable.

**Provenance and peer review** Not commissioned; externally peer reviewed.

**Data availability statement** No data are available. No original data were generated in this study.

**ORCID iDs**
Rebecca Lane Evans http://orcid.org/0000-0002-8002-9342
Chinwe Onuegbu http://orcid.org/0000-0001-6372-9390
Frances Griffiths http://orcid.org/0000-0002-4173-1438

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
