## [Reviewer comments · BMJ Open]

ARTICLE DETAILS

TITLE (PROVISIONAL)	A Systematic Review of Educational Interventions to improve the Menstrual Health of young adolescent girls.
AUTHORS	Evans, Rebecca; Harris, Bronwyn; Onuegbu, Chinwe; Griffiths, Frances

VERSION 1 – REVIEW

REVIEWER	Haque, Syed UChicago Research Bangladesh
REVIEW RETURNED	08-Nov-2021

GENERAL COMMENTS	Figure 2: 1. The reason why 852 articles were excluded? Methods: 1. The author can include the explanation why they included the literature between 2014 to 2020? Table 2:---- The table can be a summary. That will make clearer to the readers.
--

REVIEWER	Yilmaz, Siobhan K. University of New Mexico
REVIEW RETURNED	01-Dec-2021

GENERAL COMMENTS	Article Summary This study incorporates both a systematic review and meta-analysis of the small subset of quantitative studies included in the systematic review, to address the impacts of menstrual education interventions in young adolescent (inexperienced) menstruating populations. The primary outcome analyzed was changes in knowledge or skill, with secondary outcomes in some cases of changes in attitude surrounding menstruation, following the KAP framework. Effect sizes were measured using Cohen's d and a synthesis of findings was operationalized using a logic model to theorize the role of menstrual education initiatives and their subsequent impacts on menstrual health (broadly speaking). In total, this is a fairly useful piece of research which attempts to bring together multi-faceted studies towards a central theme. This study does emphasize and bring credence to the necessity and legitimacy of increasing menstruation-related curricula in schools, worldwide; however, there are several points which would make this article stand on firmer ground and read more smoothly.
--

	Minor Comments A careful read-through of the manuscript will reveal a number of sentences which do not end with a punctuation mark. There are extraneous punctuation marks, such as on pg. 16, line 36 where “(11),(12,13)” should more likely read “(11,12,13)”. On pg. 18, line 5 it would seem that a bullet point is missing before “Increase skills and competencies...” On pg. 24, when the “Quantitative Results” are first presented, the text reads that n=11 for the number of studies included in the meta-analysis, but Figure 2 list an n of 10. Which is correct? Table 3 could use some polishing so that the column titles are more legible (perhaps adjust the font size). For example, it is unclear what column 10 “Number of individuals achieving a” should read, and it is unclear why this column itself is even included. On pg. 29, the “Menstrual Attitudes” section lists five studies, but the citations in parenthesis are six – perhaps the inclusion of citation 31 is not meant to be listed here? Pg. 29, line 23-24 includes the statement “Interventions that are multi-component may be more successful than those that are single-component”. This needs to be supported by a formal citation from literature. Major Comments It would be useful to further elaborate on the decision process behind the initial screening where 852 potential studies were eliminated, given that 24 articles seems a rather small sample size when the initial screening found 900. Further, discussion of the effect sizes and meta-analysis should acknowledge the limitations of having such a small sample of quantitative studies on which to make calculations and subsequent conclusions. The final column in Table 3 lists the impact of the intervention, but the authors need to further clarify/justify upon which criteria these judgments of impact have been made, and include this in the main text of the manuscript. I am curious as to whether the younger age of the samples in the included studies was an a priori choice by the researchers, or rather was this just a function of the studies the authors were able to find through their literature search? If it was a specific choice, as would be indicated by the authors statement that they excluded studies with interventions aimed at adult women, the authors should further justify this decision in the main manuscript. While the discussion and explanation of the logic model is important to the worth of this piece, the description of it on pg. 21 seems out of place. It would likely fit better in the manuscript if that section was moved to later-on in the article and extend the portion titled “Synthesis: The Logic Model” on pg. 30. Additionally, it may be helpful to create a cleaner transition between the results of the meta-analysis/systematic review and the synthesis by way of the logic model presentation/discussion. The authors should be very clear that they are not stating that gains in knowledge/skills will directly lead to increases in access to menstrual products (see pg. 21, line 10 & pg. 30, line 35). Access to products, in of itself, is more likely driven by external factors, rather than awareness of what types of products are necessary for hygienic and healthy menstrual management.
--	--

	You have emphasized that effect sizes appear largest for the “interactive” interventions, and have briefly brought up the mechanism of constructivist theory. You may also wish to include discussion of other conceptual frameworks which also point to the benefits of including physical interaction in health/hygiene educational approaches (see: Yilmaz, SK, et al. “Touch for Health: Use of Pavlovian Processes with Physical Touch as a Means to Improve Menstrual Hygiene Management Initiatives, Measured by Willingness to Pay.” PharmacoEconomics - Open https://doi.org/10.1007/s41669-019-0168-6.) This article may also benefit from some further discussion of the limitation that only one study came from a high-income country, and yet, there is the growing recognition and consequences of “Period Poverty”, and what are the implications of this lack of coverage.
--	--

REVIEWER	Pike, Meghan Dalhousie University
REVIEW RETURNED	15-Dec-2021

GENERAL COMMENTS	December 15th, 2021 Dear Authors, Thank you for your commitment to this work: this was a large undertaking and I commend the authors for their dedication to this subject. The background is nicely written and provides an excellent overview of the history of menstrual education and the importance of stigma. I appreciate the focus on younger adolescents as there is a paucity of research on young adolescents’ experience with menstruation. Overall, the study has merit, but requires further attention prior to being considered for publication. Please see my recommendations for how the article could be strengthened. Abstract  • To me as a reader and menstrual educator, my most valuable take-away from this article is the comparison of effects of different interventions (e.g. pamphlets and books vs. peer teaching) – last paragraph under Menstrual Knowledge. Consider including this in your Abstract to strengthen it. Consider highlighting this as your key result in Conclusion and Discussion. • Comments from the PRISMA Abstract checklist:  o The exclusion criteria for the review are not specified in the Abstract o The Abstract indicates that “seven wide-ranging databases” were used to find articles; the PRISMA checklist requires that information sources used be specified. o The total number of participants are missing from results. • The Concluding paragraph needs some strengthening; the authors state: “indeed, all indicators are positive.” What is meant by this? Can you elaborate? • Strengths and Limitations are listed in bullet form: check that this is according to Author Guidelines. Methods  • There are a few items missing from the PRISMA checklist. If these elements are not included in the study, authors should describe why in a supplementary file:
--

- o 13e, 13f
- o 16b – excluded studies should be cited
- Define what you mean by “inexperienced menstruators” in the first paragraph of methods. At first I assumed this meant within a certain number of years post-menarche, but later I think you specified it is referring to girls with intellectual disability? Please clarify. Note that under “Participants” you do not include “inexperienced menstruators.” This should be consistent.
- Can you clarify how many reviewers conducted a quality assessment of each article? I think this is included in Contributions but is not clear in the methods.
- I wonder if “Synthesis: Building A Logic Model” may fit better under Results section.

Supplementary File: Characteristics and quality of included studies

- Does including intellectually-disabled girls affect the generalizability of your results? I wonder if this is a separate research question altogether. Perhaps consider presenting results separately for young adolescents and girls with intellectual disability.
- I noticed that some studies included children of both sexes, but the objective of the review specifies “young adolescent girls”; How was this accounted for when interpreting studies and synthesizing them to generalize results?
- In the footnote: it states that The Best Fit principle was used. What is The Best Fit principle? Can this be referenced?

Limitations

- Consider commenting on barriers to implementing menstrual education in developing countries – for example, would a school in a developing country have the funding to provide published books for every student? Or the resources to provide training to staff? Consider in your review how the availability of resources in a given country may impact the type of menstrual education that can be provided.
- The study period was between January 2014 – May 2020, which is prior to the COVID-19 global pandemic. Consider commenting on the affect of the global pandemic on delivering menstrual education. If, based on your review, the interactive sessions are the most effective, how do you propose educators deliver interactive sessions on menstrual education during a pandemic?
- Authors state in methods that the Cohen’s d has not been previously calculated for this discipline and therefore the magnitude of its effect size can only be considered relative to others in this review. Please consider discussing this in limitations. Researchers must be careful in conclusions if it is unknown what effect size is meaningful.

Results

- Only one study was from a high-income country. Would the study be strengthened if this study was excluded and the results could focus on low- and middle-income countries? How much did the results from the one study with a high-income country affect overall results?
- Can you give some examples as to what are considered interactive interventions (e.g. in the Abstract)?
- What does this mean: “There were four pairs of papers, each reporting” Please elaborate.

	 • Quality Assessment: How do you suggest blinding in an educational intervention? How did some studies achieve this? • Table 2: I think this can be summarized in a narrative paragraph - I don't think it matters which study employed which intervention specifically and it is an additional table/page that doesn't provide information that can't be clearly represented in a narrative - e.g. "out of X studies included for analysis, the most common intervention was X (%), followed by Y, Z, etc." • Menstrual Attitudes: Any time "significantly" or "significance" is used, please provide statistical measure of significance with a precision estimate (e.g. p-value with confidence intervals). • Menstrual Attitudes: "maybe did not resonate" – please remove this as it should be part of discussion, not included in presentation of results. • Menstrual Practice: "reducing the incidence of STIs from 19.2% to 12.9%" – was this difference statistically significant? Conclusions  • Implications for Policy and Practice: Consider including here what one should take from this article when developing a menstrual education program - i.e. strategies to make it interactive, etc. Contributors  • "RE used the MMAT to assess the quality and this was verified by FG and BH." This statement should be included clearly in the methods. Minor comments  • Check spelling of the word "focussed" versus "focused" • In general, sentences should not begin with "but." (Inclusion & Exclusion criteria, second to last paragraph). • Menstrual Practice: "culminating" – did you mean "culminated?" • Figure 2: PRISMA flow chart: What is meant by 'other sources'?
--	---

VERSION 1 – AUTHOR RESPONSE

Response to review of Systematic Review BMJ Open

Reviewer 1.

Article Summary

This study incorporates both a systematic review and meta-analysis of the small subset of quantitative studies included in the systematic review, to address the impacts of menstrual education interventions in young adolescent (inexperienced) menstruating populations. The primary outcome analyzed was changes in knowledge or skill, with secondary outcomes in some cases of changes in attitude surrounding menstruation, following the KAP framework. Effect sizes were measured using Cohen's d and a synthesis of findings was operationalized using a logic model to theorize the role of menstrual education initiatives and their subsequent impacts on menstrual health (broadly speaking). In total, this is a fairly useful piece of research which attempts to bring together multi-faceted studies towards a central theme. This study does emphasize and bring credence to the necessity and legitimacy of increasing menstruation-related curricula in schools, worldwide; however, there are several points which would make this article stand on firmer ground and read more smoothly.

Minor Comments

- **A careful read-through of the manuscript will reveal a number of sentences which do not end with a punctuation mark.**

Thank you for your attention to detail. We hope we have corrected the omissions now.

- **There are extraneous punctuation marks, such as on pg. 16, line 36 where “(11),(12,13)” should more likely read “(11,12,13)”.**

Thank you. We have re-read and checked the article for extraneous punctuation marks and hope we have corrected them now.

- **On pg. 18, line 5 it would seem that a bullet point is missing before “Increase skills and competencies...”**

Thank you. That is now corrected.

- **On pg. 24, when the “Quantitative Results” are first presented, the text reads that n=11 for the number of studies included in the meta-analysis, but Figure 2 list an n of 10. Which is correct?**

N=11. Figure 2 has been corrected. Thank -you.

- **Table 3 could use some polishing so that the column titles are more legible (perhaps adjust the font size). For example, it is unclear what column 10 “Number of individuals achieving a” should read, and it is unclear why this column itself is even included.**

Thank you. We have removed some columns and combined others to make the table clearer, and have described the pertinent columns in the results section.

- **On pg. 29, the “Menstrual Attitudes” section lists five studies, but the citations in parenthesis are six – perhaps the inclusion of citation 31 is not meant to be listed here?**

Thank you – you are correct! It has now been removed.

- **Pg. 29, line 23-24 includes the statement “Interventions that are multi-component may be more successful than those that are single-component”. This needs to be supported by a formal citation from literature.**

Thank you. We have removed this statement from the results section.

Major Comments

- **It would be useful to further elaborate on the decision process behind the initial screening where 852 potential studies were eliminated, given that 24 articles seems a rather small sample size when the initial screening found 900. Further, discussion of the effect sizes and meta-analysis should acknowledge the limitations of having such a small sample of quantitative studies on which to make calculations and subsequent conclusions.**

Thank-you for this question and the opportunity to clarify why 852 articles were excluded. These articles did not meet the inclusion criteria. There were a large number from the health care sector about menstruation that focussed on menstrual disorders and menstrual co-morbidities. They were not interventions. There were some from the WASH sector that focussed on improvements to menstrual hygiene but they did not include an educational component. A small minority were educational interventions but of the wrong age group.

This has now been changed in the text to: 852 did not meet the inclusion criteria. This was largely because they were not interventions but studies of menstruation.

- **Further, discussion of the effect sizes and meta-analysis should acknowledge the limitations of having such a small sample of quantitative studies on which to make calculations and subsequent conclusions.**

We have addressed this issue in the results section:

Due to the limited and heterogeneous nature of the data, we interpret the results only relative to the other studies in this review.

- **The final column in Table 3 lists the impact of the intervention, but the authors need to further clarify/justify upon which criteria these judgments of impact have been made, and include this in the main text of the manuscript.**

Thank you. We have remodelled this and put it into the main text.

The average effect size of studies in this review was 3.44. Taking this as a medium effect size, we ranked them lowest – highest and suggest that <2 is low and >5 is high. Where we could not calculate an effect size, we have calculated % change in score.

- **I am curious as to whether the younger age of the samples in the included studies was an a priori choice by the researchers, or rather was this just a function of the studies the authors**

were able to find through their literature search? If it was a specific choice, as would be indicated by the authors statement that they excluded studies with interventions aimed at adult women, the authors should further justify this decision in the main manuscript.

Thank you. It was an a priori decision to look at the menstrual knowledge of young adolescents girls 10-14 years so that they were of limited menstrual experience.

We have added that to the paragraph.

Studies that included adolescents up to the age of 19 were not excluded if the aim were to instruct menstruators with limited experience of menstruation. Some studies included older girls because they were intellectually disabled and were part of the intervention based on intellectual age rather than chronological age. Some studies included older girls because they were members of classes assigned by grade rather than age. Studies about interventions aimed at adult women were excluded.

While the discussion and explanation of the logic model is important to the worth of this piece, the description of it on pg. 21 seems out of place. It would likely fit better in the manuscript if that section was moved to later-on in the article and extend the portion titled “Synthesis: The Logic Model” on pg. 30. Additionally, it may be helpful to create a cleaner transition between the results of the meta-analysis/systematic review and the synthesis by way of the logic model presentation/discussion.

Thank you for your insight. We agree and have moved it to the results section so that it reads more logically and we hope there is a smoother transition to the discussion.

• The authors should be very clear that they are not stating that gains in knowledge/skills will directly lead to increases in access to menstrual products (see pg. 21, line 10 & pg. 30, line 35). Access to products, in of itself, is more likely driven by external factors, rather than awareness of what types of products are necessary for hygienic and healthy menstrual management.

Thank you

We have changed to wording to emphasize the knowledge gains rather than the access.

Below that is the proximal outcome; hygienic and comfortable menstruation management, which requires knowledge of the preparation, care and maintenance or disposal of menstrual products; the importance of keeping oneself clean and dry to avoid infection and chafing, and self-care practices to manage the symptoms of dysmenorrhea.

The indicators of proximal outcomes (that girls can manage their menstruation hygienically and comfortably) are a timely use of suitable menstrual products and painkillers; the use of water and soap to clean away menstrual blood, and self-care practices such as practising yoga or taking a rest.

• You have emphasized that effect sizes appear largest for the “interactive” interventions, and have briefly brought up the mechanism of constructivist theory. You may also which to include discussion of other conceptual frameworks which also point to the benefits of including physical interaction in health/hygiene educational approaches (see: Yilmaz, SK, et al. “Touch for Health: Use of Pavlovian Processes with Physical Touch as a Means to Improve Menstrual Hygiene Management Initiatives, Measured by Willingness to Pay.” *PharmacoEconomics - Open* <https://doi.org/10.1007/s41669-019-0168-6>.)

Thank-you for drawing our attention to this. It is very useful and has been included in the discussion as suggested.

Those interventions that demonstrated skills and allowed for physical touch were also very effective.

Other hygiene interventions that have been evaluated have pointed to the positive impact of a physical interaction with the tools of behaviour change (69).

• This article may also benefit from some further discussion of the limitation that only one study came from a high-income country, and yet, there is the growing recognition and consequences of “Period Poverty”, and what are the implications of this lack of coverage

Thank you for all of your useful comments. We have tried to address the issues you have identified.

1. Figure 2:

The reason why 852 articles were excluded?

Thank-you for this question and the opportunity to clarify why 852 articles were excluded. These articles did not meet the inclusion criteria. There were a large number from the health care sector about menstruation that focussed on menstrual disorders and menstrual co-morbidities. They were not interventions. There were some from the WASH sector that focussed on improvements to menstrual hygiene but they did not include an educational component. A small minority were educational interventions but of the wrong age group.

This has now been changed in the text to: 852 did not meet the inclusion criteria. This was largely because they were not interventions but studies of menstruation.

2. Methods:

The author can include the explanation why they included the literature between 2014 to 2020?

Thank you for the opportunity to explain why the date of January 2014 was chosen. This is because a systematic review by Hennegan and Montgomery (2016) included articles up until January 2015. It was also at this time that the language moved from Menstrual Hygiene Management to Menstrual Health and this review was intended to capture articles that reflected this change in emphasis, by starting in January 2014.

The paragraph at the end of the introduction has been altered to

Two previous reviews of papers published prior to January 2015 focussed on the more narrow 'Menstrual Hygiene Management' in LMIC (34,35). The term Menstrual Health is now preferred to hygiene, partly to avoid the suggestion that menstruation *per se* is unclean, but mostly to emphasize the holistic nature of the menstrual experience. This literature review only includes publications since 2014 in order to capture that change.

3. Table 2:---- The table can be a summary. That will make clearer to the readers.

Thank you for your comment. We have decided to remove the table. The contents are available in the table in appendix 2.

Reviewer 2

Thank you for your very encouraging comments.

1. Abstract

- To me as a reader and menstrual educator, my most valuable take-away from this article is the comparison of effects of different interventions (e.g. pamphlets and books vs. peer teaching) – last paragraph under Menstrual Knowledge. Consider including this in your Abstract to strengthen it. Consider highlighting this as your key result in Conclusion and Discussion.

Thank you so much for highlighting this as the take-away message. We have changed the results and conclusions of the abstract to emphasize this and hopefully make it clearer.

'Results

The meta-analysis indicates that larger effect sizes were attained with the more interactive interventions such as peer-teaching. The lowest effect sizes were attained by those that distributed pamphlets or leaflets. Where measured, confidence also improved.

Conclusions

Education interventions are effective in increasing the menstrual knowledge of young adolescent girls and skills training improves competency to manage menstruation more hygienically and comfortably.

Interactive interventions that encouraged discussion were more effective than those in which the information was didactic or written. Girls gain confidence when they can discuss what is normal, share coping strategies and receive emotional support.'

- 2. • Comments from the PRISMA Abstract checklist:
 - o The exclusion criteria for the review are not specified in the Abstract
- This has now been included:

Eligibility criteria

Interventions in which there was a component of menstrual information transfer were included. Interventions which had not been evaluated, or studies of menstruation which were not interventions were excluded.

Comment: Consider rephrase: Studies of menstruation that did not include evaluation of an intervention were excluded.

- o The Abstract indicates that "seven wide-ranging databases" were used to find articles; the PRISMA checklist requires that information sources used be specified.

The list of data bases has now been transferred to the Abstract

ASSIA Applied Social Science Index and Abstracts; CINAHL Cumulative Index to Nursing and Allied Health Literature; EMBASE Excerpta Medica database; MEDLINE Medical Literature Analysis and Retrieval System Online; Sociological Abstracts; Web of Science; IBBS International Bibliography of the Social Sciences; TRoPHI Trials Register of Promoting Health Interventions.

- o The total number of participants are missing from results.

Apologies for this omission. It has now been added to the results.

The number of participants varied from 1 to 2564. The total number of participants was 10362.

Comment: I think you can just comment on the total number of participants for the Abstract.

- The Concluding paragraph needs some strengthening; the authors state: "indeed, all indicators are positive." What is meant by this? Can you elaborate?

Thank you for questioning this. We have followed your advice to focus on the importance of the type of intervention (see above) and we have removed this sentence.

- Strengths and Limitations are listed in bullet form: check that this is according to Author Guidelines. Yes thank you for querying this. We want to get it right! This is what the author guidelines say, so we have left the bullet points untouched.

- **Please include a 'Strengths and limitations of this study' section after the abstract.** This section should be no more than 5 bullet points relating specifically to the methods – not the results of the study. This will be published as a summary box after the abstract in the final published article.

3. Methods

- There are a few items missing from the PRISMA checklist. If these elements are not included in the study, authors should describe why in a supplementary file:
 - o 13e, 13f

Unfortunately the studies were very heterogeneous and further analyses or modelling was not possible.

A meta-analysis was conducted on 11 studies which measured a change in menstrual knowledge following an intervention. A visual inspection of forest plots showed that all studies found a significant improvement in menstrual knowledge. Where studies reported the mean and standard

deviation of a menstrual knowledge questionnaire, we calculated the effect size (Cohen's d). Due to the limited and heterogenous nature of the data, further statistical analyses were not possible.

o 16b – excluded studies should be cited

Papers that appeared to meet the criteria but were eventually excluded were by Bhagwat et al 2020(40), as the participants were the wrong age group; Mastorci et al 2019(41) and Tuli et al 2019 (42) as these were not interventions.

Comment: Now that you have clearly defined your exclusion criteria, I do not think either of these need to be cited ☺ You can remove unless you think these studies needed further justification as to why they were excluded (i.e. were excluded for a unique reason rather than fitting the exclusion criteria).

- Define what you mean by “inexperienced menstruators” in the first paragraph of methods. At first I assumed this meant within a certain number of years post-menarche, but later I think you specified it is referring to girls with intellectual disability? Please clarify. Note that under “Participants” you do not include “inexperienced menstruators.” This should be consistent.

Yes, thank you for pointing out these inconsistencies. This phrase has been changed to adolescents ‘with limited experience of menstruation’ and is written as such in the Abstract and the Methods. We also added it to the inclusion criteria where we gave more detail on the inclusion of intellectually-disabled teenagers. Two studies did report on girls with an intellectual disability.

‘Studies that included adolescents up to the age of 19 were not excluded if the aim were to instruct menstruators with limited experience of menstruation. Some studies included older girls because they were intellectually disabled and were part of the intervention based on intellectual age rather than chronological age. Some studies included older girls because they were members of classes assigned by grade rather than age. But studies about interventions aimed at adult women were excluded.’

- Can you clarify how many reviewers conducted a quality assessment of each article? I think this is included in Contributions but is not clear in the methods.

Thank you for your attention to detail. We have added further clarification as it appeared in contributions.

Two reviewers (RE and CO) screened abstracts and full texts of all citations obtained for eligibility independently. Data extraction of eligible material and the quality assessment was conducted by RE using a data extraction framework agreed upon by FG, BH and RE. The quality assessment tool was agreed upon by RE, FG and BH. RE used the MMAT to assess the quality and this was verified by FG and BH.

- I wonder if “Synthesis: Building A Logic Model” may fit better under Results section.

Thank you for your suggestion. The Logic Model section has now been moved to the results.

4. Supplementary File: Characteristics and quality of included studies

- Does including intellectually-disabled girls affect the generalizability of your results? I wonder if this is a separate research question altogether. Perhaps consider presenting results separately for young adolescents and girls with intellectual disability.

Comment: I don't think this comment has been addressed. Since there were only 2 studies including adolescents with intellectual disability, I no longer think you need to present results

separately. However, I would consider adding this as a potential limitation as your study as it does contribute to heterogeneity of the included studies. In addition, this may be a very interesting area for future research – i.e. how do we provide effective menstrual education to adolescents with intellectual disabilities (exciting idea!!)

- I noticed that some studies included children of both sexes, but the objective of the review specifies “young adolescent girls”; How was this accounted for when interpreting studies and synthesizing them to generalize results?

Thank you for asking about this. We were primarily looking for types of menstrual education and this included puberty lessons delivered to boys as well as girls. In our objectives we refer to ‘young adolescents’ and we have tried to clarify this by adding to the inclusion criteria:

‘Interventions that were straightforward ‘Menstrual Education’ in which lessons about puberty, anatomy, and hygiene were delivered by teachers or other educators, to both boys and girls, were eligible.’

We used the results from a study including boys to conclude that experience was important in menstrual attitude, and have added a line to highlight that point.

‘The only intervention which did not find a significant difference in attitude pre- and post-test involved puberty education videos shown to early adolescent boys and girls (47) and maybe, because of lack of experience, did not resonate so much.’

Comment: Consider rephrase: “...shown to early adolescent boys and girls. It is possible that this intervention did not resonate with the boys, who lack personal experience with menstruation.” Something like that?

- In the footnote: it states that The Best Fit principle was used. What is The Best Fit principle? Can this be referenced?

Thank you. The procedure has now been clarified and referenced.

‘Framework analysis and the ‘best fit’ principle were used to score the studies (Carroll, Booth and Cooper, 2011; Suto, 2012; Carroll *et al.*, 2013). All studies were interrogated with two questions ‘Are there clear research questions?’ and ‘Do the collected data allow the research questions to be addressed?’ which were considered fundamental to the quality and were scored on a scale of ‘Yes’ = 2, ‘not clear’ = 1 and ‘No’ = 0. Five further supplementary questions were considered that addressed quality issues such as sample size. The sets of questions were different depending upon the study design, and are not directly comparable, so less weight was given to these; Yes = 1 and No = 0. The maximum score when added together was 2 + 2 + 5 = 9. Studies scored 0-5 were categorised as low quality (as it was possible to get these scores without clear research questions or valid methods); those that scored 6 or 7 were scored as moderate quality and those that scored 8 or 9 were scored as high quality (a subjective scale based on personal expertise (Suto, 2012) and community of practice validation (Bejar, 2011)).’

Limitations

- Consider commenting on barriers to implementing menstrual education in developing countries – for example, would a school in a developing country have the funding to provide published books for every student? Or the resources to provide training to staff? Consider in your review how the

availability of resources in a given country may impact the type of menstrual education that can be provided.

We take your point and have rewritten the paragraph on Implications for policy and practice.

'This review provides evidence that menstrual education has a positive effect on the menstrual knowledge, attitudes and practices of adolescent girls and needs to be delivered by trained personnel who can facilitate discussion. However, without an enabling infrastructure, the progress towards menstrual health will be limited. In order to achieve the more distal outcomes of the logic model, programme and policy makers need to address the menstrual literacy of the wider population. It would be prudent to ensure that menstrual education is embedded into school curricula. . Especially but not exclusively in LMIC, where resources are limited, it would be prudent to ensure that menstrual education is embedded into school and the training of teachers would be a good start.'

- The study period was between January 2014 – May 2020, which is prior to the COVID-19 global pandemic. Consider commenting on the affect of the global pandemic on delivering menstrual education. If, based on your review, the interactive sessions are the most effective, how do you propose educators deliver interactive sessions on menstrual education during a pandemic?

This is a very good point. Thank you. We have added this paragraph to implications for policy and practice.

This review was carried out on studies conducted before the Global Pandemic began in March 2020. It has had a profound effect on education, and many programmes have had to go on-line. We would encourage Menstrual educators to be mindful of the benefits of interaction, and make use of on-line teaching platforms that facilitate discussion in break-out rooms.

- Authors state in methods that the Cohen's d has not been previously calculated for this discipline and therefore the magnitude of its effect size can only be considered relative to others in this review. Please consider discussing this in limitations. Researchers must be careful in conclusions if it is unknown what effect size is meaningful.

Agreed. We have repeated a comment that was in the methods and put it into the limitations.

Cohen's d has not previously been calculated for this discipline and therefore the magnitude of the effect size can only be considered relative to others in this review.

Results

- Only one study was from a high-income country. Would the study be strengthened if this study was excluded and the results could focus on low- and middle-income countries? How much did the results from the one study with a high-income country affect overall results?

Thank-you. These are important considerations. We wanted to have as broad interpretation of menstrual education as possible, and it was surprising that there was only one study from a high-income country. We don't believe that excluding it alters the results particularly; indeed, it's effect size seems to fit the pattern. The overall conclusions about interactive education are generalisable to both HIC and LMIC, so we have decided to leave it in.

- Can you give some examples as to what are considered interactive interventions (e.g. in the Abstract)?

Yes, thank you. We have added

'with the more interactive interventions such as peer-teaching. The lowest effect sizes were attained by those that distributed pamphlets or leaflets.'

- What does this mean: "There were four pairs of papers, each reporting" Please elaborate. Thank you – we realise that this adds nothing to the review and it has been deleted.

- Quality Assessment: How do you suggest blinding in an educational intervention? How did some studies achieve this? Thank you; yes it is of course very difficult to blind in an educational intervention. It is usually done at the whole-school level. Some of the RCT's did randomly assign whole schools within districts to intervention and control.

We changed the paragraph to refer to this.

The methodological quality of study designs was mixed: Eleven were rated as high quality and thirteen as moderate to low. Those considered to be of the highest quality were randomized controlled trials which included comparison groups. Some of the studies (nine) did this at the whole-school level which is recommended in educational interventions to prevent contamination of the intervention group with the control (57). The research questions were clear and the data collection methods appropriate. Of the other studies, several methodological limitations were noted; commonly, neither the delivery team nor the participants were blinded (nine); adequate randomization of the participants was lacking (ten) and /or relevant confounds were not identified or controlled (four). The quality of data analysis also varied considerably, with the weakest having small sample sizes and no measure of statistical significance (two).

- Table 2: I think this can be summarized in a narrative paragraph - I don't think it matters which study employed which intervention specifically and it is an additional table/page that doesn't provide information that can't be clearly represented in a narrative - e.g. "out of X studies included for analysis, the most common intervention was X (%), followed by Y, Z, etc."

Thank you; we agree that the information in the table is superfluous and have removed it. The information can still be found in appendix 2.

- Menstrual Attitudes: Any time "significantly" or "significance" is used, please provide statistical measure of significance with a precision estimate (e.g. p-value with confidence intervals).

Thank you – we have added the figures as reported by the studies.

Four interventions reported significantly different ($p < 0.05$) attitude scores, and three of those provided pamphlets that addressed cultural restrictions(4,50,54). The other was an intervention on dysmenorrhea and self-care included pamphlets with video and peer-sharing and girls who had taken part had a significant increase in confidence and decrease ($p < 0.001$) in 'bothersome' menstrual

attitude (52). The only intervention which did not find a significant difference in attitude pre- and post-test involved puberty education videos shown to early adolescent boys and girls (43).

- Menstrual Attitudes: “maybe did not resonate” – please remove this as it should be part of discussion, not included in presentation of results.

Thank you, we have removed it.

- Menstrual Practice: “reducing the incidence of STIs from 19.2% to 12.9%” – was this difference statistically significant?

Thank you – yes this was significant. We have added the p-value.

‘A feasibility trial into the use of the menstrual cup by school girls in Kenya (49) found that usage increased as time went on and culminating in 96% usage after 9 months. There was also an increase in hygiene, with the menstrual cup reported as reducing the prevalence of STIs from 19.2% to 12.9% (p=0.018) (62).’

Conclusions

- Implications for Policy and Practice: Consider including here what one should take from this article when developing a menstrual education program - i.e. strategies to make it interactive, etc.

We have written the implications to include your suggestion. Thank you.

‘This review provides evidence that menstrual education has a positive effect on the menstrual knowledge, attitudes and practices of adolescent girls and needs to be delivered by trained personnel who are confident to lead discussion. Especially but not exclusively in LMIC, where resources are limited, it would be prudent to ensure that menstrual education is embedded into the school curriculum and that teachers receive specialist training.

Progress towards Menstrual Health is limited without an enabling environment. In order to achieve the more distal outcomes of the logic model, programme and policy makers need to address the menstrual literacy of the wider population. Multi-component interventions that speak to different actors and include hardware and software provision alongside menstrual education may make Menstrual Health more attainable.’

Contributors

- “RE used the MMAT to assess the quality and this was verified by FG and BH.” This statement should be included clearly in the methods.

Thank you. This is now in methods.

Minor comments

- Check spelling of the word “focussed” versus “focused”

We have used the UK spelling.

- In general, sentences should not begin with “but.” (Inclusion & Exclusion criteria, second to last paragraph).

This has been removed. Thank you.

- Menstrual Practice: “culminating” – did you mean “culminated?”

Thank you we have changed it to culminated.

- Figure 2: PRISMA flow chart: What is meant by ‘other sources’?

Thank you. This refers to the possibility of identifying records within the grey literature, particularly the reports of NGOs who carry out interventions.

Review 3

VERSION 2 – REVIEW

REVIEWER	Yilmaz, Siobhan K. University of New Mexico
REVIEW RETURNED	24-Feb-2022

GENERAL COMMENTS	The efforts that the authors have taken to improve the flow and completeness of the work are appreciated.
---

REVIEWER	Pike, Meghan Dalhousie University
REVIEW RETURNED	14-Mar-2022

GENERAL COMMENTS	Response to review of Systematic Review BMJ Open Reviewer 1. Article Summary This study incorporates both a systematic review and meta-analysis of the small subset of quantitative studies included in the systematic review, to address the impacts of menstrual education interventions in young adolescent (inexperienced) menstruating populations. The primary outcome analyzed was changes in knowledge or skill, with secondary outcomes in some cases of changes in attitude surrounding menstruation, following the KAP framework. Effect sizes were measured using Cohen’s d and a synthesis of findings was operationalized using a logic model to theorize the role of menstrual education initiatives and their subsequent impacts on menstrual health (broadly speaking). In total, this is a fairly useful piece of research which attempts to bring together multi-faceted studies towards a central theme. This study does emphasize and bring credence to the necessity and legitimacy of increasing menstruation-related curricula in schools, worldwide; however, there are several points which would make this article stand on firmer ground and read more smoothly. Minor Comments [ ] A careful read-through of the manuscript will reveal a number of sentences which do not end with a punctuation mark. Thank you for your attention to detail. We hope we have corrected the omissions now.
---

There are extraneous punctuation marks, such as on pg. 16, line 36 where “(11),(12,13)” should more likely read “(11,12,13)”.

Thank you. We have re-read and checked the article for extraneous punctuation marks and hope we have corrected them now.

On pg. 18, line 5 it would seem that a bullet point is missing before “Increase skills and competencies...”

Thank you. That is now corrected.

On pg. 24, when the “Quantitative Results” are first presented, the text reads that n=11 for the number of studies included in the meta-analysis, but Figure 2 list an n of 10. Which is correct?

N=11. Figure 2 has been corrected. Thank -you.

Table 3 could use some polishing so that the column titles are more legible (perhaps adjust the font size). For example, it is unclear what column 10 “Number of individuals achieving a” should read, and it is unclear why this column itself is even included.

Thank you. We have removed some columns and combined others to make the table clearer, and have described the pertinent columns in the results section.

On pg. 29, the “Menstrual Attitudes” section lists five studies, but the citations in parenthesis are six – perhaps the inclusion of citation 31 is not meant to be listed here?

Thank you – you are correct! It has now been removed.

Pg. 29, line 23-24 includes the statement “Interventions that are multi-component may be more successful than those that are single-component”. This needs to be supported by a formal citation from literature.

Thank you. We have removed this statement from the results section.

Major Comments

It would be useful to further elaborate on the decision process behind the initial screening where 852 potential studies were eliminated, given that 24 articles seems a rather small sample size when the initial screening found 900. Further, discussion of the effect sizes and meta-analysis should acknowledge the limitations of having such a small sample of quantitative studies on which to make calculations and subsequent conclusions.

Thank-you for this question and the opportunity to clarify why 852 articles were excluded. These articles did not meet the inclusion criteria. There were a large number from the health care sector about menstruation that focussed on menstrual disorders and menstrual co-morbidities. They were not interventions. There were some from the WASH sector that focussed on improvements to menstrual hygiene but they did not include an educational

component. A small minority were educational interventions but of the wrong age group.

This has now been changed in the text to: 852 did not meet the inclusion criteria. This was largely because they were not interventions but studies of menstruation.

Further, discussion of the effect sizes and meta-analysis should acknowledge the limitations of having such a small sample of quantitative studies on which to make calculations and subsequent conclusions.

We have addressed this issue in the results section:

Due to the limited and heterogeneous nature of the data, we interpret the results only relative to the other studies in this review.

The final column in Table 3 lists the impact of the intervention, but the authors need to further clarify/justify upon which criteria these judgments of impact have been made, and include this in the main text of the manuscript.

Thank you. We have remodelled this and put it into the main text.

The average effect size of studies in this review was 3.44. Taking this as a medium effect size, we ranked them lowest – highest and suggest that <2 is low and >5 is high. Where we could not calculate an effect size, we have calculated % change in score.

I am curious as to whether the younger age of the samples in the included studies was an a priori choice by the researchers, or rather was this just a function of the studies the authors were able to find through their literature search? If it was a specific choice, as would be indicated by the authors statement that they excluded studies with interventions aimed at adult women, the authors should further justify this decision in the main manuscript.

Thank you. It was an a priori decision to look at the menstrual knowledge of young adolescents girls 10-14 years so that they were of limited menstrual experience.

We have added that to the paragraph.

Studies that included adolescents up to the age of 19 were not excluded if the aim were to instruct menstruators with limited experience of menstruation. Some studies included older girls because they were intellectually disabled and were part of the intervention based on intellectual age rather than chronological age. Some studies included older girls because they were members of classes assigned by grade rather than age. Studies about interventions aimed at adult women were excluded.

While the discussion and explanation of the logic model is important to the worth of this piece, the description of it on pg. 21 seems out of place. It would likely fit better in the manuscript if that section was moved to later-on in the article and extend the portion titled "Synthesis: The Logic Model" on pg. 30. Additionally, it may be helpful to create a cleaner transition between the results of the

meta-analysis/systematic review and the synthesis by way of the logic model presentation/discussion.

Thank you for your insight. We agree and have moved it to the results section so that it reads more logically and we hope there is a smoother transition to the discussion.

□ The authors should be very clear that they are not stating that gains in knowledge/skills will directly lead to increases in access to menstrual products (see pg. 21, line 10 & pg. 30, line 35). Access to products, in of itself, is more likely driven by external factors, rather than awareness of what types of products are necessary for hygienic and healthy menstrual management.

Thank you

We have changed to wording to emphasize the knowledge gains rather than the access.

Below that is the proximal outcome; hygienic and comfortable menstruation management, which requires knowledge of the preparation, care and maintenance or disposal of menstrual products; the importance of keeping oneself clean and dry to avoid infection and chafing, and self-care practices to manage the symptoms of dysmenorrhea.

The indicators of proximal outcomes (that girls can manage their menstruation hygienically and comfortably) are a timely use of suitable menstrual products and painkillers; the use of water and soap to clean away menstrual blood, and self-care practices such as practising yoga or taking a rest.

□ You have emphasized that effect sizes appear largest for the “interactive” interventions, and have briefly brought up the mechanism of constructivist theory. You may also which to include discussion of other conceptual frameworks which also point to the benefits of including physical interaction in health/hygiene educational approaches (see: Yilmaz, SK, et al. “Touch for Health: Use of Pavlovian Processes with Physical Touch as a Means to Improve Menstrual Hygiene Management Initiatives, Measured by Willingness to Pay.” *PharmacoEconomics - Open* <https://doi.org/10.1007/s41669-019-0168-6>.)

Thank-you for drawing our attention to this. It is very useful and has been included in the discussion as suggested.

Those interventions that demonstrated skills and allowed for physical touch were also very effective. Other hygiene interventions that have been evaluated have pointed to the positive impact of a physical interaction with the tools of behaviour change (69).

□ This article may also benefit from some further discussion of the limitation that only one study came from a high-income country, and yet, there is the growing recognition and consequences of “Period Poverty”, and what are the implications of this lack of coverage

Thank you for all of your useful comments. We have tried to address the issues you have identified.

1. Figure 2:

The reason why 852 articles were excluded?

Thank-you for this question and the opportunity to clarify why 852 articles were excluded. These articles did not meet the inclusion criteria. There were a large number from the health care sector about menstruation that focussed on menstrual disorders and menstrual co-morbidities. They were not interventions. There were some from the WASH sector that focussed on improvements to menstrual hygiene but they did not include an educational component. A small minority were educational interventions but of the wrong age group.

This has now been changed in the text to: 852 did not meet the inclusion criteria. This was largely because they were not interventions but studies of menstruation.

2. Methods:

The author can include the explanation why they included the literature between 2014 to 2020?

Thank you for the opportunity to explain why the date of January 2014 was chosen. This is because a systematic review by Hennegan and Montgomery (2016) included articles up until January 2015. It was also at this time that the language moved from Menstrual Hygiene Management to Menstrual Health and this review was intended to capture articles that reflected this change in emphasis, by starting in January 2014.

The paragraph at the end of the introduction has been altered to

Two previous reviews of papers published prior to January 2015 focussed on the more narrow 'Menstrual Hygiene Management' in LMIC (34,35). The term Menstrual Health is now preferred to hygiene, partly to avoid the suggestion that menstruation per se is unclean, but mostly to emphasize the holistic nature of the menstrual experience. This literature review only includes publications since 2014 in order to capture that change.

3. Table 2:---- The table can be a summary. That will make clearer to the readers.

Thank you for your comment. We have decided to remove the table. The contents are available in the table in appendix 2.

Reviewer 2

Thank you for your very encouraging comments.

1. Abstract

□ To me as a reader and menstrual educator, my most valuable take-away from this article is the comparison of effects of different interventions (e.g. pamphlets and books vs. peer teaching) – last paragraph under Menstrual Knowledge. Consider including this in your Abstract to strengthen it. Consider highlighting this as your key result in Conclusion and Discussion.

Thank you so much for highlighting this as the take-away message. We have changed the results and conclusions of the abstract to emphasize this and hopefully make it clearer.

'Results

The meta-analysis indicates that larger effect sizes were attained with the more interactive interventions such as peer-teaching. The lowest effect sizes were attained by those that distributed pamphlets or leaflets. Where measured, confidence also improved.

Conclusions

Education interventions are effective in increasing the menstrual knowledge of young adolescent girls and skills training improves competency to manage menstruation more hygienically and comfortably. Interactive interventions that encouraged discussion were more effective than those in which the information was didactic or written. Girls gain confidence when they can discuss what is normal, share coping strategies and receive emotional support.'

2. □ Comments from the PRISMA Abstract checklist:

o The exclusion criteria for the review are not specified in the Abstract

This has now been included:

Eligibility criteria

Interventions in which there was a component of menstrual information transfer were included. Interventions which had not been evaluated, or studies of menstruation which were not interventions were excluded.

	Comment: Consider rephrase: Studies of menstruation that did not include evaluation of an intervention were excluded. o The Abstract indicates that “seven wide-ranging databases” were used to find articles; the PRISMA checklist requires that information sources used be specified. The list of data bases has now been transferred to the Abstract ASSIA Applied Social Science Index and Abstracts; CINAHL Cumulative Index to Nursing and Allied Health Literature; EMBASE Excerpta Medica database; MEDLINE Medical Literature Analysis and Retrieval System Online; Sociological Abstracts; Web of Science; IBBS International Bibliography of the Social Sciences; TRoPHI Trials Register of Promoting Health Interventions. o The total number of participants are missing from results. Apologies for this omission. It has now been added to the results. The number of participants varied from 1 to 2564. The total number of participants was 10362. Comment: I think you can just comment on the total number of participants for the Abstract. [ ] The Concluding paragraph needs some strengthening; the authors state: “indeed, all indicators are positive.” What is meant by this? Can you elaborate? Thank you for questioning this. We have followed your advice to focus on the importance of the type of intervention (see above) and we have removed this sentence. [ ] Strengths and Limitations are listed in bullet form: check that this is according to Author Guidelines. Yes thank you for querying this. We want to get it right! This is what the author guidelines say, so we have left the bullet points untouched. Please include a ‘Strengths and limitations of this study’ section after the abstract. This section should be no more than 5 bullet points relating specifically to the methods – not the results of the study. This will be published as a summary box after the abstract in the final published article. 3. Methods
--	--

There are a few items missing from the PRISMA checklist. If these elements are not included in the study, authors should describe why in a supplementary file:

o 13e, 13f

Unfortunately the studies were very heterogeneous and further analyses or modelling was not possible.

A meta-analysis was conducted on 11 studies which measured a change in menstrual knowledge following an intervention. A visual inspection of forest plots showed that all studies found a significant improvement in menstrual knowledge. Where studies reported the mean and standard deviation of a menstrual knowledge questionnaire, we calculated the effect size (Cohen's d). Due to the limited and heterogeneous nature of the data, further statistical analyses were not possible.

o 16b – excluded studies should be cited

Papers that appeared to meet the criteria but were eventually excluded were by Bhagwat et al 2020(40), as the participants were the wrong age group; Mastorci et al 2019(41) and Tuli et al 2019 (42) as these were not interventions.

Comment: Now that you have clearly defined your exclusion criteria, I do not think either of these need to be cited You can remove unless you think these studies needed further justification as to why they were excluded (i.e. were excluded for a unique reason rather than fitting the exclusion criteria).

Define what you mean by “inexperienced menstruators” in the first paragraph of methods. At first I assumed this meant within a certain number of years post-menarche, but later I think you specified it is referring to girls with intellectual disability? Please clarify. Note that under “Participants” you do not include “inexperienced menstruators.” This should be consistent.

Yes, thank you for pointing out these inconsistencies. This phrase has been changed to adolescents ‘with limited experience of menstruation’ and is written as such in the Abstract and the Methods.

We also added it to the inclusion criteria where we gave more detail on the inclusion of intellectually-disabled teenagers. Two studies did report on girls with an intellectual disability.

'Studies that included adolescents up to the age of 19 were not excluded if the aim were to instruct menstruators with limited experience of menstruation. Some studies included older girls because they were intellectually disabled and were part of the intervention based on intellectual age rather than chronological age. Some studies included older girls because they were members of classes assigned by grade rather than age. But studies about interventions aimed at adult women were excluded.'

Can you clarify how many reviewers conducted a quality assessment of each article? I think this is included in Contributions but is not clear in the methods.

Thank you for your attention to detail. We have added further clarification as it appeared in contributions.

Two reviewers (RE and CO) screened abstracts and full texts of all citations obtained for eligibility independently. Data extraction of eligible material and the quality assessment was conducted by RE using a data extraction framework agreed upon by FG, BH and RE. The quality assessment tool was agreed upon by RE, FG and BH. RE used the MMAT to assess the quality and this was verified by FG and BH.

I wonder if "Synthesis: Building A Logic Model" may fit better under Results section.

Thank you for your suggestion. The Logic Model section has now been moved to the results.

4. Supplementary File: Characteristics and quality of included studies

- Does including intellectually-disabled girls affect the generalizability of your results? I wonder if this is a separate research question altogether. Perhaps consider presenting results separately for young adolescents and girls with intellectual disability.

Comment: I don't think this comment has been addressed. Since there were only 2 studies including adolescents with intellectual disability, I no longer think you need to present results separately. However, I would consider adding this as a potential limitation as your study as it does contribute to heterogeneity of the included studies. In addition, this may be a very interesting area for future research – i.e. how do we provide effective menstrual education to adolescents with intellectual disabilities (exciting idea!!)

• I noticed that some studies included children of both sexes, but the objective of the review specifies “young adolescent girls”; How was this accounted for when interpreting studies and synthesizing them to generalize results?

Thank you for asking about this. We were primarily looking for types of menstrual education and this included puberty lessons delivered to boys as well as girls. In our objectives we refer to ‘young adolescents’ and we have tried to clarify this by adding to the inclusion criteria:

‘Interventions that were straightforward ‘Menstrual Education’ in which lessons about puberty, anatomy, and hygiene were delivered by teachers or other educators, to both boys and girls, were eligible.’

We used the results from a study including boys to conclude that experience was important in menstrual attitude, and have added a line to highlight that point.

‘The only intervention which did not find a significant difference in attitude pre- and post-test involved puberty education videos shown to early adolescent boys and girls (47) and maybe, because of lack of experience, did not resonate so much.’

Comment: Consider rephrase: “...shown to early adolescent boys and girls. It is possible that this intervention did not resonate with the boys, who lack personal experience with menstruation.”
Something like that?

• In the footnote: it states that The Best Fit principle was used. What is The Best Fit principle? Can this be referenced?

Thank you. The procedure has now been clarified and referenced.

‘Framework analysis and the ‘best fit’ principle were used to score the studies (Carroll, Booth and Cooper, 2011; Suto, 2012; Carroll et al., 2013). All studies were interrogated with two questions ‘Are there clear research questions?’ and ‘Do the collected data allow the research questions to be addressed?’ which were considered fundamental to the quality and were scored on a scale of ‘Yes’ = 2, ‘not clear’ = 1 and ‘No’ = 0. Five further supplementary questions

were considered that addressed quality issues such as sample size. The sets of questions were different depending upon the study design, and are not directly comparable, so less weight was given to these; Yes = 1 and No = 0. The maximum score when added together was 2 + 2 + 5 = 9. Studies scored 0-5 were categorised as low quality (as it was possible to get these scores without clear research questions or valid methods); those that scored 6 or 7 were scored as moderate quality and those that scored 8 or 9 were scored as high quality (a subjective scale based on personal expertise (Suto, 2012) and community of practice validation (Bejar, 2011).’

Limitations

- Consider commenting on barriers to implementing menstrual education in developing countries – for example, would a school in a developing country have the funding to provide published books for every student? Or the resources to provide training to staff? Consider in your review how the availability of resources in a given country may impact the type of menstrual education that can be provided.

We take your point and have rewritten the paragraph on Implications for policy and practice.

‘This review provides evidence that menstrual education has a positive effect on the menstrual knowledge, attitudes and practices of adolescent girls and needs to be delivered by trained personnel who can facilitate discussion. However, without an enabling infrastructure, the progress towards menstrual health will be limited. In order to achieve the more distal outcomes of the logic model, programme and policy makers need to address the menstrual literacy of the wider population. It would be prudent to ensure that menstrual education is embedded into school curricula. . Especially but not exclusively in LMIC, where resources are limited, it would be prudent to ensure that menstrual education is embedded into school and the training of teachers would be a good start.’

- The study period was between January 2014 – May 2020, which is prior to the COVID-19 global pandemic. Consider commenting on the affect of the global pandemic on delivering menstrual education. If, based on your review, the interactive sessions are the most effective, how do you propose educators deliver interactive sessions on menstrual education during a pandemic?

This is a very good point. Thank you. We have added this paragraph to implications for policy and practice.

	This review was carried out on studies conducted before the Global Pandemic began in March 2020. It has had a profound effect on education, and many programmes have had to go on-line. We would encourage Menstrual educators to be mindful of the benefits of interaction, and make use of on-line teaching platforms that facilitate discussion in break-out rooms.  • Authors state in methods that the Cohen's d has not been previously calculated for this discipline and therefore the magnitude of its effect size can only be considered relative to others in this review. Please consider discussing this in limitations. Researchers must be careful in conclusions if it is unknown what effect size is meaningful. Agreed. We have repeated a comment that was in the methods and put it into the limitations. Cohen's d has not previously been calculated for this discipline and therefore the magnitude of the effect size can only be considered relative to others in this review. Results  • Only one study was from a high-income country. Would the study be strengthened if this study was excluded and the results could focus on low- and middle-income countries? How much did the results from the one study with a high-income country affect overall results? Thank-you. These are important considerations. We wanted to have as broad interpretation of menstrual education as possible, and it was surprising that there was only one study from a high-income country. We don't believe that excluding it alters the results particularly; indeed, it's effect size seems to fit the pattern. The overall conclusions about interactive education are generalisable to both HIC and LMIC, so we have decided to leave it in.  • Can you give some examples as to what are considered interactive interventions (e.g. in the Abstract)? Yes, thank you. We have added
--	--

'with the more interactive interventions such as peer-teaching. The lowest effect sizes were attained by those that distributed pamphlets or leaflets.'

- What does this mean: "There were four pairs of papers, each reporting" Please elaborate.

Thank you – we realise that this adds nothing to the review and it has been deleted.

- Quality Assessment: How do you suggest blinding in an educational intervention? How did some studies achieve this? Thank you; yes it is of course very difficult to blind in an educational intervention. It is usually done at the whole-school level. Some of the RCT's did randomly assign whole schools within districts to intervention and control.

We changed the paragraph to refer to this.

The methodological quality of study designs was mixed: Eleven were rated as high quality and thirteen as moderate to low. Those considered to be of the highest quality were randomized controlled trials which included comparison groups. Some of the studies (nine) did this at the whole-school level which is recommended in educational interventions to prevent contamination of the intervention group with the control (57). The research questions were clear and the data collection methods appropriate. Of the other studies, several methodological limitations were noted; commonly, neither the delivery team nor the participants were blinded (nine); adequate randomization of the participants was lacking (ten) and /or relevant confounds were not identified or controlled (four). The quality of data analysis also varied considerably, with the weakest having small sample sizes and no measure of statistical significance (two).

- Table 2: I think this can be summarized in a narrative paragraph - I don't think it matters which study employed which intervention specifically and it is an additional table/page that doesn't provide information that can't be clearly represented in a narrative - e.g. "out of X studies included for analysis, the most common intervention was X (%), followed by Y, Z, etc."

Thank you; we agree that the information in the table is superfluous and have removed it. The information can still be found in appendix 2.

- Menstrual Attitudes: Any time “significantly” or “significance” is used, please provide statistical measure of significance with a precision estimate (e.g. p-value with confidence intervals).

Thank you – we have added the figures as reported by the studies.

Four interventions reported significantly different ($p < 0.05$) attitude scores, and three of those provided pamphlets that addressed cultural restrictions(4,50,54). The other was an intervention on dysmenorrhea and self-care included pamphlets with video and peer-sharing and girls who had taken part had a significant increase in confidence and decrease ($p < 0.001$) in ‘bothersome’ menstrual attitude (52). The only intervention which did not find a significant difference in attitude pre- and post-test involved puberty education videos shown to early adolescent boys and girls (43).

- Menstrual Attitudes: “maybe did not resonate” – please remove this as it should be part of discussion, not included in presentation of results.

Thank you, we have removed it.

- Menstrual Practice: “reducing the incidence of STIs from 19.2% to 12.9%” – was this difference statistically significant?

Thank you – yes this was significant. We have added the p-value.

‘A feasibility trial into the use of the menstrual cup by school girls in Kenya (49) found that usage increased as time went on and culminating in 96% usage after 9 months. There was also an increase in hygiene, with the menstrual cup reported as reducing the prevalence of STIs from 19.2% to 12.9% ($p = 0.018$) (62).’

Conclusions

• Implications for Policy and Practice: Consider including here what one should take from this article when developing a menstrual education program - i.e. strategies to make it interactive, etc.

We have written the implications to include your suggestion. Thank you.

‘This review provides evidence that menstrual education has a positive effect on the menstrual knowledge, attitudes and practices of adolescent girls and needs to be delivered by trained personnel who are confident to lead discussion. Especially but not exclusively in LMIC, where resources are limited, it would be prudent to ensure that menstrual education is embedded into the school curriculum and that teachers receive specialist training.

Progress towards Menstrual Health is limited without an enabling environment. In order to achieve the more distal outcomes of the logic model, programme and policy makers need to address the menstrual literacy of the wider population. Multi-component interventions that speak to different actors and include hardware and software provision alongside menstrual education may make Menstrual Health more attainable.’

Contributors

• “RE used the MMAT to assess the quality and this was verified by FG and BH.” This statement should be included clearly in the methods.

Thank you. This is now in methods.

Minor comments

• Check spelling of the word “focussed” versus “focused”

We have used the UK spelling.

• In general, sentences should not begin with “but.” (Inclusion & Exclusion criteria, second to last paragraph).

This has been removed. Thank you.

• Menstrual Practice: “culminating” – did you mean “culminated?”

Thank you we have changed it to culminated.

• Figure 2: PRISMA flow chart: What is meant by ‘other sources’?

Thank you. This refers to the possibility of identifying records within the grey literature, particularly the reports of NGOs who carry out interventions.

	Review 3
--	----------

VERSION 2 – AUTHOR RESPONSE

Thank you for your comments, which are very useful. We have made the corrections suggested.